# Characteristics of Age Classification into Five-Year Intervals to Explain Sarcopenia and Immune Cells in Older Adults

**DOI:** 10.3390/medicina59101700

**Published:** 2023-09-22

**Authors:** Seung-Jae Heo, Yong-Seok Jee

**Affiliations:** 1Department of Life Sports Education, Kongju National University, Gongju-si 32588, Republic of Korea; 2Research Institute of Sports and Industry Science, Hanseo University, Seosan-si 31962, Republic of Korea

**Keywords:** sarcopenia, calf circumference, grip strength, appendicular skeletal muscle mass, natural killer cell, immune cells

## Abstract

*Background and Objectives*: This study focused on investigating sarcopenic factors and immune cells in older adulthood. To achieve this, the variables related to sarcopenia and immune cells in people living in the same community were analyzed. *Materials and Methods*: A total of 433 elderly individuals aged 61 to 85 years were randomly categorized as follows in 5-year intervals: 68 in the youngest-old group (aged 61–65), 168 in the young-old group (aged 66–70), 127 in the middle-old group (aged 71–75), 46 in the old-old group (aged 76–80), and 19 in the oldest-old group (aged 81–85). *Results*: With the progression of age, calf circumference (−8.4 to −11.05%; *p* = 0.001) and grip strength (−9.32 to −21.01%; *p* = 0.001) exhibited a noticeable reduction with each successive 5-year age bracket. Conversely, the capability to complete the five-time chair stand demonstrated a clear incline (32.49 to 56.81%; *p* = 0.001), starting from the middle-aged group. As for appendicular skeletal muscle mass, there was an evident tendency for it to decrease (−7.08 to −26.62%; *p* = 0.001) with increasing age. A gradual decline in natural killer cells became apparent within the old-old and oldest-old groups (−9.28 to −26.27%; *p* = 0.001). The results of the post hoc test revealed that CD3 T cells showcased their peak levels in both the youngest-old and young-old groups. This was followed by the middle-old and old-old groups, with slightly lower levels. This pattern was similarly observed in CD4 T cells, CD8 T cells, and CD19 B cells. *Conclusions*: This study reaffirmed that sarcopenia and immune cell function decline with each successive 5-year increase in age. Considering these findings, the importance of implementing programs aimed at ensuring a high-quality extension of life for the elderly is strongly underscored.

## 1. Introduction

By the year 2030, it is estimated that the number of people aged 60 years and older will reach 1.4 billion, and by 2050, it is expected to further grow to 2.1 billion [1]. Because of this aging population, there is a growing need for policies and initiatives that cater to the specific needs and challenges faced by older adults to ensure their well-being and quality of life [2]; this is viewed as a significant step towards prioritizing efforts to promote the well-being and health of older adults worldwide [3].

Adjusting age group categorization is one way to ensure that healthcare and research efforts align with the specific needs and characteristics of older individuals within distinct age ranges [4]. Given the wide range of health conditions and the rising prevalence of chronic diseases, it can be recommended that most research employing 5-year age intervals may yield beneficial outcomes [5]. By using 5-year age intervals, researchers and healthcare practitioners can better account for the emergence of various chronic conditions and the potential differences in health status within specific age ranges. For instance, considering the survival rate of cancer survivors is essential, as it can impact the health status and needs of older adults within specific age brackets to improve the health condition of older adults [6].

Sarcopenia, which involves the loss of muscle mass and strength, can significantly reduce quality of life if left untreated or poorly managed. Sarcopenia can lead to mobility issues and increased dependency, which can adversely affect an individual’s physical fitness and performance. Addressing sarcopenia and other age-related health conditions becomes crucial to mitigate their negative impact on quality of life [7]. Understanding the factors contributing to sarcopenia’s progression can pave the way for the development of more effective preventive and treatment strategies. By taking proactive measures and intervening early, healthcare providers can potentially reduce the burden of sarcopenia and enhance the quality of life of older adults, promoting healthier and more fulfilling lives as people age. Sarcopenia can be categorized as a geriatric syndrome, which is defined by the Asian Working Group for Sarcopenia (AWGS) as an “age-related loss of skeletal muscle mass plus loss of muscle strength and/or reduced physical performance” [7,8]. It is also considered a generalized and progressive skeletal muscle disorder that is associated with an increased likelihood of adverse outcomes by the European Working Group on Sarcopenia in Older People 2 (EWGSOP2) [9]. In particular, the decrease in muscle mass that occurs with sarcopenia is accompanied by a decline in immune cell function, leading to an increased susceptibility to various infections [10].

The immune system has been suggested to play a role in affecting myogenesis, as indicated in many studies [11,12]. While significant progress has been made in understanding the interaction between the skeletal muscle and the immune system, the impact of health status within specific age ranges on this relationship remains relatively unexplored. Further research in this area could provide valuable insights into the underlying mechanisms of sarcopenia and the immune system in aged adults. Consequently, it is essential to investigate variations in sarcopenia and immunocytes based on various age categorizations. Such detailed age-specific categorization can make it easier to prevent muscle atrophy in the elderly and can be considered a proactive measure to prevent the decline in immune cell function that comes with aging. This can contribute to maintaining a high quality of life and extending the lifespan. Therefore, the primary objective of this study was to examine and compare the characteristics of sarcopenia and immunocytes by categorizing the elderly into age groups with intervals of 5 years.

## 2. Materials and Methods

### 2.1. Study Design and Participants

This research took place in June and July 2023. This single-blind, randomized, prospective cohort study was conducted in a research center at Seoul Seniors Towers, Korea. Through promotional activities at the Seoul Senior Tower, individuals who voluntarily enlisted in the study were expected to be between the ages of 60 and 100 and capable of independent living. Participants were unaware of the randomization results. The variables to be investigated in this study were determined at the time of analysis. The randomization list was generated by a computer using block randomization with five blocks. Prior to the study, the principal investigator explained all the procedures to the participants in detail. All participants read and signed an informed consent form. This study followed the principles of the Declaration of Helsinki and received approval from the institutional ethics committee (HS23-08-02, 2 November 2022 to 1 November 2023). This study was also registered with the Clinical Research Information Service, reference KCT0008712. This study determined the required sample size for the ANOVA (fixed effects, omnibus, one-way) using G*Power software (ver. 3.1.9.7; Heinrich-Heine-Universität Düsseldorf, Düsseldorf, Germany). The levels of sarcopenia factors and immune cells were then examined. The inclusion criteria comprised individuals without cancer or severe illnesses, those capable of performing a certain level of physical activity, and individuals who had not undergone surgery within the past 3 months. Older adults with severe musculoskeletal disorders, lung disease, cancer, and cachexia, which could potentially influence muscle mass and function, were excluded from the study [13].

After eliminating 139 individuals from the initial pool of 572 eligible participants, the remaining 433 participants (males, 53.5%; females, 46.5%) were distributed across five distinct groups. Among the 69 individuals belonging to the youngest-old category (aged 61–65 years), 1 person was unavailable during the follow-up stage. Within the young-old group (aged 66–70 years), consisting of 169 individuals, 1 participant was also lost during the follow-up phase. In the middle-old group, which consisted of 128 individuals aged 71–75 years, 1 individual was also not included in the follow-up phase. Notably, there were no dropouts in the old-old group (aged 76–80 years), which consisted of 46 individuals. Among the 21 individuals in the oldest-old group (aged 81–85 years), 2 participants were lost during the follow-up phase, as illustrated in Figure 1.

### 2.2. Daily Physical Activity Measurements

The participants’ daily physical activity levels were assessed and quantified using the International Physical Activity Questionnaire (IPAQ)—shortened form version [14]. Participants filled out questionnaires based on their weekly physical activity records, and daily calorie expenditure was determined by calculating metabolic equivalent minutes.

### 2.3. Anthropometric Measurements

All participants underwent an assessment of their height, body weight, body mass index (BMI), lean mass, and fat mass. These measurements were conducted using a body composition analyzer known as the Inbody 770 (Biospace Co., Ltd., Seoul, Republic of Korea). This analyzer employs the bioelectrical impedance analysis (BIA) technique to offer insight into the distribution of various tissues within the body.

### 2.4. Sarcopenia Measures

Following the AWGS guidelines [7], the first step involved evaluating calf circumference. The participants’ left knees were bent to achieve a 90-degree angle, and a measuring tape was placed around the calf area. The tape was then gently moved along the calf to identify the point of greatest circumference, with care taken to avoid compressing the subcutaneous tissue [15]. The calf circumference values were collected from both sides and averaged to establish the ultimate calf circumference measurement. The established cutoff values were <34 cm for males and <33 cm for females. Next, the assessment involved measuring grip strength, accomplished using a grip dynamometer (TKK 5401; Takei, Tokyo, Japan). Participants were instructed to stand with their feet at shoulder width and comfortably adjust the dynamometer to fit their hand’s length, and then squeeze it with their maximum force. Each participant’s right and left hands underwent two tests each, and the average of these values was employed for subsequent analysis. The established cutoff values for assessing sarcopenia based on grip strength were <28 kg for males and <18 kg for females, as indicated by previous studies [7,16]. In the third step, the participants’ physical performance was gauged through the five-time chair stand test. In this assessment, participants were required to rise from a chair five times sequentially, with their arms folded across their chest. The time taken to complete the task was recorded in seconds [16]. The established diagnostic criterion for identifying sarcopenia using this test is a duration of ≥12 s for both males and females, as outlined in prior research [7]. The dual-energy X-ray absorptiometry (DXA) technique (Lunar Co., Madison, WI, USA), was employed to determine the appendicular skeletal muscle (ASM) mass and subsequently diagnose sarcopenia. In this procedure, elderly participants assumed a comfortable lying position, and a full-body scan was conducted. The scanning process generally spanned around 15 to 20 min. To compute the ASM mass using DXA, the muscle masses from both the arms and legs were aggregated and subsequently divided by the square of the participant’s height in meters. The established criteria for diagnosing sarcopenia based on ASM mass are 7 kg/m^2^ or below for males and 5.4 kg/m^2^ or below for females [7].

### 2.5. Immunocyte Measures

The blood collection and analysis process in the study involved several steps and utilized specific tubes and techniques for different measurements. (1) Blood Collection: Participants underwent a fasting period of 10 h before blood samples were collected from the median cubital vein. (2) Differential Blood Cell Counts. (3) Flow Cytometric Assays: Flow cytometry was performed to assess cellular immune function. (4) Serum Sample Processing: After collection, the serum samples were allowed to clot for 30–45 min in an upright position; they were then subjected to centrifugation to separate the serum from the blood cells. (5) Quantification of Immunocytes: To evaluate cellular immune function, the quantification of immunocytes was performed using flow cytometry; specific antibodies were used to target different cell types, including CD56+ NK cells, CD3+ T-cells, CD4+ helper T-cells, CD8+ cytotoxic T-cells, and CD19+ B-cells. The use of specific tubes and techniques ensured accurate and reliable measurements of different parameters [17]. Natural killer cells have the ability to identify infected or cancerous cells, binding to them, and then releasing enzymes and other substances that effectively break down the outer membrane of these cells [18]. These cells play a significant role in the early defense against viral infections. At birth, acquired immunity is not present; instead, it develops through a learning process. This learning process begins when an individual’s immune system encounters a foreign intruder and identifies its antigen. Subsequently, the adaptive immune components learn the most effective way to combat this antigen and commence building a memory for it [19]. Acquired immunity is also known as specific immunity because it orchestrates a tailored response against specific antigens encountered in the past. As we grow older, our immune system tends to lose some of its ability to differentiate between self and non-self. Consequently, autoimmune diseases may manifest more frequently. For instance, the quantity of white blood cells and their subsets capable of responding to new antigens diminishes, which, in turn, diminishes the body’s capacity to retain memory and mount a defense when encountering new antigens [20].

### 2.6. Data Analyses

The data were analyzed using SPSS software (version 25; IBM Corp., Armonk, NY, USA). Graphical representations of the results were generated using GraphPad Prism 10.0 (La Jolla, CA, USA). Descriptive statistics, such as mean and standard deviation, were used to summarize the data. Categorical variables were presented as counts (*n*) and percentages (%) and compared using the χ^2^ test. The Shapiro–Wilk test was utilized to evaluate the regularity of anthropometric and clinical measurements. The equality of variances was assessed, and the mean values were analyzed using one-way ANOVA when assuming equal variances. In cases where equal variances were not assumed, the Kruskal–Wallis test was employed. Additionally, ANCOVA verification was performed to assess the factors related to sarcopenia and immune cell variables with respect to age. To identify noteworthy protocol impacts, the Scheffe test was performed for further analysis. In addition, this study calculated delta values to assess differences between the youngest-old and young-old groups, between the young-old and middle-old groups, between the middle-old and old-old groups, and between the old-old and oldest-old groups, respectively. Furthermore, this study conducted an analysis using the Kruskal–Wallis test to examine intergroup differences in these delta values. Effect sizes were determined by converting partial eta-squared to Cohen’s d [21]; values were classified as small (0.00 ≤ d ≤ 0.49), medium (0.50 ≤ d ≤ 0.79), and large (d ≤ 0.80). A significance level of *p* ≤ 0.05 was used to determine significance.

## 3. Results

### 3.1. Demographic, Anthropometric, and Clinical Characteristics

As indicated in Table 1, there was a notable variation in the age of the participants across the groups. In the youngest-old category group, there were 44 males and 24 females. The young-old group had 89 males and 79 females. Within the middle-old group, there were 62 males and 65 females. The old-old group had 25 males and 21 females. Among the 19 individuals in the oldest-old group, there were 9 males and 10 females. The gender distribution did not display any significant divergence among the five groups. In a characteristic manner, lean mass exhibited a decline with increasing age, whereas fat mass demonstrated no distinction between the youngest-old and young-old groups. Subsequently, there was an upswing in fat mass after the middle-old group.

The older adults who took part in this study commonly shared conditions such as overweight, obesity, diabetes mellitus, hypertension, hyperlipidemia, arthritis, and low back pain. Regarding clinical traits, only hypertension, arthritis, and low back pain displayed noteworthy variations between the groups. In contrast, no substantial disparities were observed between the groups in relation to obesity, diabetes, and hyperlipidemia. The outcomes of the post hoc test revealed that hypertension was least prevalent within the youngest-old group and that it displayed a gradual escalation with advancing age. This trend indicated a rise in the frequency of hypertension cases following the old-old group. With the progression of age, there is a corresponding increase in the occurrence of low back pain among elderly individuals, with the highest prevalence observed within the oldest-old group. In terms of daily calorie expenditure from physical activities, the average calories burned through these activities were as follows: 1740.35 ± 289.28 MET·min/week for the youngest-old group, 1627.90 ± 233.20 MET·min/week for the young-old group, 1343.36 ± 187.45 MET·min/week for the middle-old group, 1211.28 ± 289.28 MET·min/week for the old-old group, and 1132.58 ± 208.81 MET·min/week for the oldest-old group. Significant differences were observed among these groups (*X*^2^ = 181.708; *p* = 0.001; η^2^ = 0.422).

### 3.2. Comparative Results of Sarcopenic Factors

As indicated in Table 2, every assessed parameter associated with sarcopenia exhibited notable distinctions across all five groups. Notably, there was a discernible decline in calf circumference and grip strength with each successive 5-year increment in age. Conversely, the ability to perform the five-time chair stand displayed evident deterioration from the middle-aged group onwards. Regarding ASM, there was a noticeable tendency for it to diminish with advancing age. Importantly, a significant reduction was observed, starting from the young-old group and beyond. When analyzing calf circumference with age as a covariate, there was no significant difference observed among the groups (F = 1.743; *p* = 0.140). However, handgrip strength exhibited a significant difference among the groups (F = 7.891; *p* = 0.001). When analyzing the five-time chair stand with age as a covariate, a significant difference was observed among the groups (F = 2.763; *p* = 0.027); similarly, the ASM also displayed a significant difference among the groups (F = 36.898; *p* = 0.001).

After post hoc analysis, calf circumference was the thickest in the youngest-old group and gradually decreased significantly with increasing age groups (Figure 2A). Grip strength also showed similar results to calf circumference (Figure 2B). On the other hand, the five-time chair stand showed the opposite results to calf circumference, being fastest in the youngest-old group (Figure 2C). ASM was highest in the youngest-old group, followed by the young-old group, and there were no significant differences among the middle-old, old-old, and oldest-old groups (Figure 2D).

In detail, the calf circumference showed differences between the youngest-old and young-old groups, with a decrease of −8.58%. Similarly, there was a −8.40% difference between the young-old and middle-old groups, a −11.05% difference between the middle-old and old-old groups, and a −9.27% difference between the old-old and oldest-old groups. These differences were statistically significant (Z = 140.957, *p* = 0.001, η^2^ = 0.357). Grip strength exhibited variations across age groups: there was a −20.36% decline between the youngest-old and young-old groups, a −9.32% difference between the young-old and middle-old groups, a −21.01% difference between the middle-old and old-old groups, and a −16.72% difference between the old-old and oldest-old groups. These disparities were statistically significant (Z = 142.088, *p* = 0.001, η^2^ = 0.348). The performance in the five-time chair stand test displayed variations among age groups: there was a 32.49% increase between the youngest-old and young-old groups, a 56.81% difference between the young-old and middle-old groups, an 18.95% difference between the middle-old and old-old groups, and a 27.75% difference between the old-old and oldest-old groups. These differences were statistically significant (Z = 319.322, *p* = 0.001, η^2^ = 0.723). The ASM exhibited disparities among age groups: there was a −26.62% decrease between the youngest-old and young-old groups, a −15.19% difference between the young-old and middle-old groups, a 0.33% difference between the middle-old and old-old groups, and a −7.08% difference between the old-old and oldest-old groups. These differences were statistically significant (Z = 176.718, *p* = 0.001, η^2^ = 0.488).

### 3.3. Comparative Results of Lymphocyte Subsets

As depicted in Table 3, the white blood cell count, serving as an indicator of immune cells, did not exhibit noteworthy variances between the groups. Nevertheless, significant distinctions were observed among the groups in relation to NK cells, as well as CD3, CD4, and CD8 T cell markers. Additionally, there were discernible variations in CD19, a marker for B cells, across the groups. When analyzing NK cells with age as a covariate, a significant difference was observed among the groups (F = 5.630; *p* = 0.001). Similarly, when analyzing CD3 T cells with age as a covariate, a significant difference was found among the groups (F = 3.796; *p* = 0.005), and CD4 T cells also exhibited a significant difference among the groups (F = 7.688; *p* = 0.001). Similarly, CD8 T cells (F = 7.947; *p* = 0.001) and CD9 B cells (F = 9.565; *p* = 0.001) also displayed significant differences among the groups.

Following the post hoc test, it was established that NK cells were most abundant within the youngest-old group. Notably, there was no discernible distinction in NK cell levels between the young-old and middle-old groups. Subsequently, a gradual decrease in NK cell levels was evident in the old-old and oldest-old groups (Figure 3). In detail, the NK cells showed variations across age groups: there was a −22.95% decline between the youngest-old and young-old groups, a −21.79% difference between the young-old and middle-old groups, a −26.27% difference between the middle-old and old-old groups, and a −9.28% difference between the old-old and oldest-old groups. These differences were statistically significant (Z = 159.849, *p* = 0.001, η^2^ = 0.371).

The outcomes of the post hoc test revealed that CD3 T cells exhibited their highest levels in both the youngest-old and young-old groups. Subsequently, the middle-old and old-old groups followed suit with slightly lower levels. The lowest levels of CD3 T cells were observed in the oldest-old group (Figure 4A). This tendency was similarly observed in CD4 T cells (Figure 4B), CD8 T cells (Figure 4C), and CD19 B cells (Figure 4D).

In detail, the CD3 T cells displayed differences across age groups: there was a −4.07% decrease between the youngest-old and young-old groups, a −7.13% difference between the young-old and middle-old groups, a −2.78% difference between the middle-old and old-old groups, and a −11.95% difference between the old-old and oldest-old groups. These disparities were statistically significant (Z = 68.006, *p* = 0.001, η^2^ = 0.170). The CD4 T cells exhibited disparities among age groups: there was a −3.08% decrease between the youngest-old and young-old groups, a −15.15% difference between the young-old and middle-old groups, a −15.45% difference between the middle-old and old-old groups, and a −3.24% difference between the old-old and oldest-old groups. These differences were statistically significant (Z = 119.355, *p* = 0.001, η^2^ = 0.274). The CD8 T cells displayed differences across age groups: there was a −18.43% decrease between the youngest-old and young-old groups, an −11.61% difference between the young-old and middle-old groups, a −19.64% difference between the middle-old and old-old groups, and an −11.83% difference between the old-old and oldest-old groups. These differences were statistically significant (Z = 135.165, *p* = 0.001, η^2^ = 0.328). The CD19 B cells showed differences across age groups: there was a −3.89% decrease between the youngest-old and young-old groups, a −20.04% difference between the young-old and middle-old groups, a −9.26% difference between the middle-old and old-old groups, and a −16.55% difference between the old-old and oldest-old groups. These differences were statistically significant (Z = 112.429, *p* = 0.001, η^2^ = 0.253).

## 4. Discussion

This study effectively identified distinct alterations in both sarcopenia-related factors and immune cell functionalities when individuals aged 61 to 85 were categorized into five-year groups. The findings revealed that lean mass, as measured by BIA, displayed no significant differences among the youngest-old, young-old, middle-old, and old-old groups. However, a notable decrease was evident after the age of 81 and above. Conversely, fat mass exhibited an increase in all groups older than the middle-old group. On a different note, concerning clinical conditions, there was a noticeable decline in the health status of individuals as age advanced. Specifically, hypertension, arthritis, and low back pain exhibited clear signs of deterioration. Furthermore, it was verified that the daily physical activity levels of the elderly progressively declined with advancing age.

Among the various factors associated with sarcopenia that were examined in this study, both calf circumference and the ability to perform the five-time chair stand consistently exhibited a decline with each successive 5-year age increment. Conversely, grip strength demonstrated a steady reduction after the group reached the age of 66 years or older. Meanwhile, there was a general tendency for ASM levels to decrease as age advanced. Notably, no significant distinction was found between the middle-old and old-old groups in terms of ASM. However, a significant and rapid decrease in ASM became evident after reaching the age of 81.

The prevalence of sarcopenia varies based on the way it is defined in the literature. According to different definitions, the occurrence rate is noted as 5–13% in individuals aged 60–70, whereas it can range from 11% to as high as 50% in those who are over 80 years old [22]. According to the EWGSOP, sarcopenia is described as a prevalent, complex, and costly health condition in the elderly population. Additionally, this geriatric syndrome is attributed to the incomplete interaction between diseases and aging across various systems, resulting in the manifestation of a constellation of signs and symptoms [8,23]. As observed in this study, an increase in age was associated with a clinically deteriorating phenomenon, and the worsening of chronic conditions and physical activity levels aligned with several preceding studies [18,19].

Multiple mechanisms potentially contribute to the initiation and advancement of sarcopenia [24]. These mechanisms encompass various factors, including protein synthesis, proteolysis, neuromuscular integrity, and muscle fat content. In individuals with sarcopenia, a combination of these mechanisms could play a role, with their relative impacts potentially changing over time. Identifying these mechanisms and their root causes is anticipated to aid in the development of intervention studies aiming to address one or more fundamental mechanisms. The concept of staging sarcopenia, which indicates the level of severity of the condition, offers valuable guidance for clinical management. EWGSOP proposes a conceptual staging framework comprising “pre-sarcopenia”, “sarcopenia”, and “severe sarcopenia [25]”. These categorizations imply that the more they are tailored to specific age groups, the more precise they become. The pre-sarcopenia stage is characterized by reduced muscle mass without a noticeable impact on muscle strength or physical performance. This stage can only be identified accurately using methods that measure muscle mass in comparison to established standard populations. The “sarcopenia” stage involves both low muscle mass and either low muscle strength or poor physical performance. The “severe sarcopenia” stage is diagnosed when all three criteria of the definition are met, namely, low muscle mass, low muscle strength, and poor physical performance [9]. Recognizing the different stages of sarcopenia could aid in selecting appropriate treatments and setting realistic recovery objectives. Staging might also facilitate the design of research studies that focus on specific stages or changes in stages over time. Comparing the findings of this study with those of previous research, it can generally be observed that before the age of 70, individuals are in a phase of pre-sarcopenia, and between the ages of 70 and 80, they seem to enter the stage of sarcopenia. However, from the age of 81 onwards, it could be interpreted that individuals transition into a more severe stage of sarcopenia characterized by significant muscle loss.

Similar to the changes in factors contributing to muscle loss and its progression, this study observed that the immune status of older individuals tends to worsen or decline in function as age increases [12]. The immune function plays a vital role as one of the body’s defense systems, protecting the body from invading bacteria and antigens [26]. The proper functioning of the immune system within the body is determined, to a large extent, by the presence of a sufficient number of robust immune cells [27]. In this study, the average cell count of white blood cells did not show significant differences with increasing age. However, innate immune cells (NK cells) and adaptive immune cells exhibited a tendency to decrease with age. These changes in the immune system appear to undergo a distinct shift around the age of 70. This age group corresponds to the middle-old cohort examined in this study, specifically individuals aged 71–75 years old. These results are consistent with studies indicating a reduction in the number of peripheral blood lymphocytes with aging [28,29], suggesting that lymphocyte subtypes are sensitive to the effects of aging. It should be noted, however, that there are also studies suggesting no change in lymphocyte count with age [28], indicating that further research is needed in the future. Consequently, age-related dysregulation and senescence of the immune system could potentially contribute to the advancement and deterioration of sarcopenia [29,30]. The immune system may participate in regulating skeletal muscle growth and regeneration in instances of acute and chronic muscle injuries [19,31]. Therefore, it is plausible to suggest that changes in the immune system associated with aging might play a significant role in the progression of sarcopenia.

In essence, this study demonstrated a consistent trend of decline in both sarcopenia-related factors and immune cell functionality as age progressed beyond 61 years. Furthermore, these findings were notably associated with the reduction in lean mass observed in clinical characteristics. It is reasonable to anticipate that implementing strategies to prevent the decline in muscle mass as individuals enter old age after the age of 61 can contribute to maintaining a high quality of life and preserving immune cell function. For example, as individuals age, it is essential to examine their susceptibility to easily occurring infections and their responsiveness to vaccinations [32]. Additionally, it is crucial to identify and manage the underlying causes of various age-related diseases, such as obesity, hypertension, atherosclerosis, osteoporosis, diabetes, and cancer [33,34]. Moreover, maintaining regular exercise habits [18], adopting a healthy diet, ensuring proper sleep patterns, and managing stress are all important measures to prevent the deterioration of immune cell function [17,35]. Ultimately, this study discovered that with advancing age, muscle mass decreases, evaluation parameters related to muscle loss factors worsen, and consequently, immune cell function deteriorates. Furthermore, it was determined that these findings were more pronounced when age was categorized into 5-year intervals. However, this study had a small sample size due to the limited number of older adults in the studied community in Korea. Furthermore, the fact that only elderly individuals residing in Korea were included as the study population could serve as a limitation.

## 5. Conclusions

In this study, when age was categorized into 5-year intervals, the researcher was able to observe a distinct pattern of muscle atrophy, and at the same time, the researcher confirmed a decline in the functions of immune cells. To prevent muscle atrophy in the elderly and observe declines in immune cell function, it is advisable to use 5-year age intervals, and through this approach, we can anticipate a higher quality of life.

## Figures and Tables

**Figure 1 medicina-59-01700-f001:**
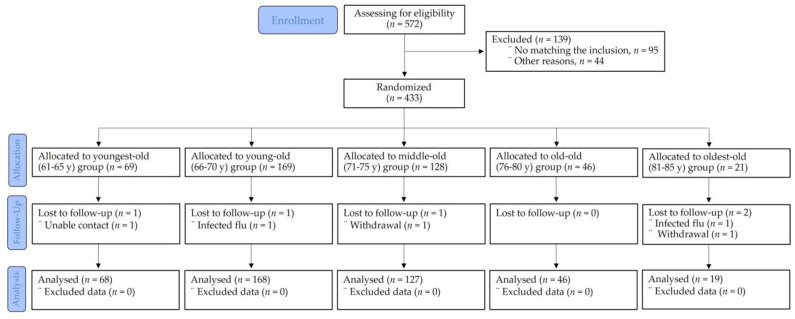
Participants’ allocation (consolidated standards for reporting of trials flow diagram).

**Figure 2 medicina-59-01700-f002:**
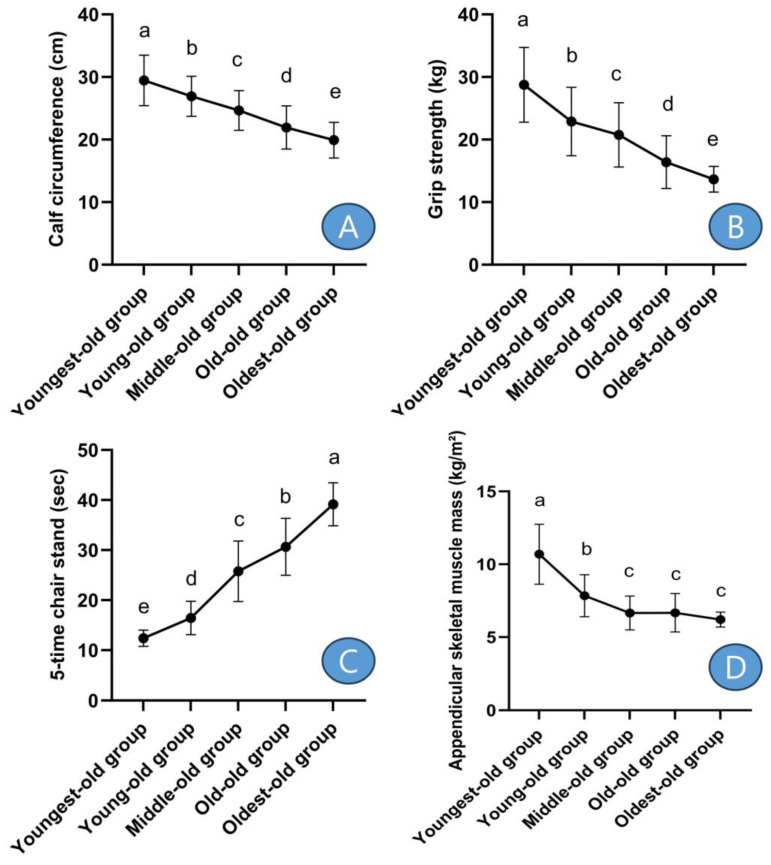
Post-hoc results of sarcopenic factors. Symbols ^a, b, c, d,^ and ^e^ represent post hoc results from the Scheffe test. (**A**) means ‘calf circumference’, (**B**) means ‘grip strength’, (**C**) means ‘5-time chair stand’, and (**D**) means ‘appendicular skeletal muscle mass’.

**Figure 3 medicina-59-01700-f003:**
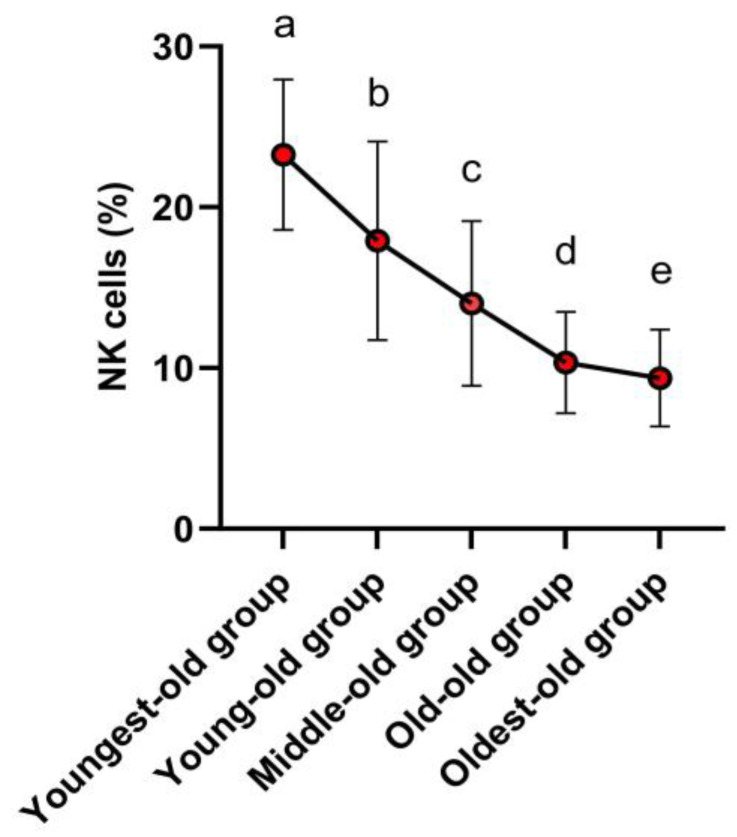
Post-hoc results of innate immune cells. Symbols ^a, b, c, d,^ and ^e^ represent post hoc results from the Scheffe test.

**Figure 4 medicina-59-01700-f004:**
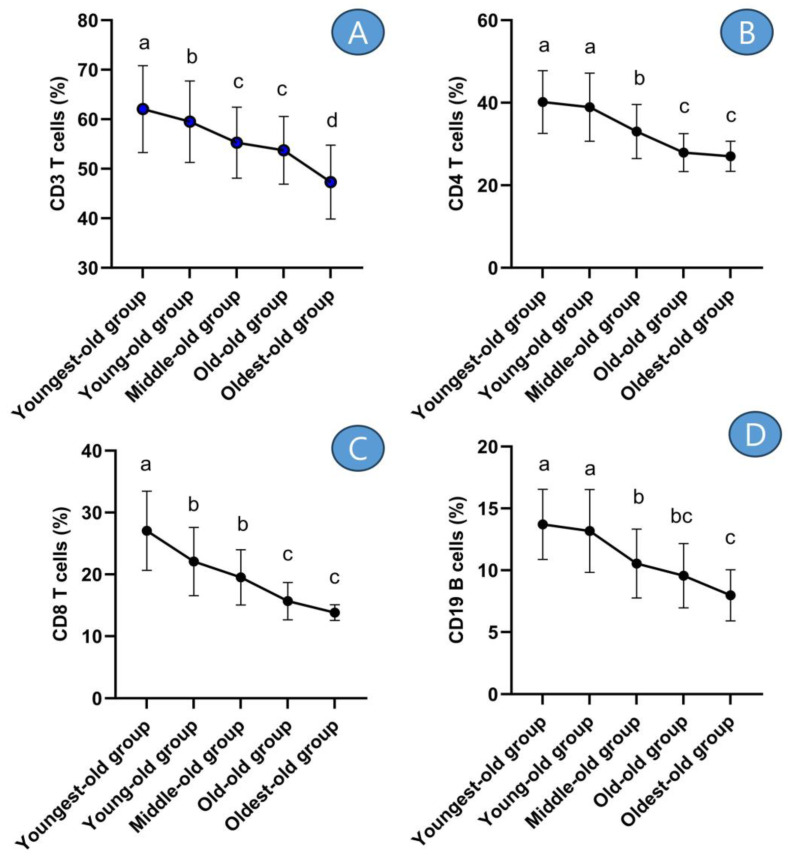
Post-hoc results of adaptive immune cells. Symbols ^a, b, c,^ and ^d^ represent post hoc results from the Scheffe test. (**A**) means ‘CD3 T cells’, (**B**) means ‘CD4 T cells’, (**C**) means ‘CD8 T cells’, and (**D**) means ‘CD19 B cells’.

**Table 1 medicina-59-01700-t001:** Demographic, anthropometric, and clinical characteristics of the old population.

	Groups					*p*	η^2^
	Youngest-Old	Young-Old	Middle-Old	Old-Old	Oldest-Old		
Items	(*n* = 68)	(*n* = 168)	(*n* = 127)	(*n* = 46)	(*n* = 19)		
Age (y)	64.46 ± 1.11	67.57 ± 1.32	73.17 ± 1.38	77.61 ± 1.27	83.00 ± 1.53	0.001	0.933
Height (cm)	170.18 ± 4.30	169.32 ± 4.30	168.85 ± 5.00	167.85 ± 4.73	167.01 ± 4.13	0.003	0.039
Body weight (kg)	70.60 ± 6.37	73.84 ± 7.66	73.24 ± 7.96	73.55 ± 7.28	69.29 ± 8.04	0.009	0.032
BMI (kg/m^2^)	24.39 ± 2.23	25.79 ± 2.85	26.07 ± 3.39	25.84 ± 2.80	24.87 ± 3.06	0.002	0.038
Lean mass (kg)	48.65 ± 6.08	47.25 ± 5.21	47.20 ± 5.85	45.65 ± 4.36	41.71 ± 4.57	0.001	0.061
Fat mass (kg)	22.98 ± 3.46	22.60 ± 2.82	24.39 ± 3.32	26.37 ± 2.82	26.21 ± 3.16	0.001	0.156
OV/OB	0.44 ± 0.50	0.49 ± 0.50	0.55 ± 0.50	0.61 ± 0.49	0.63 ± 0.50	0.285	0.012
DM	0.18 ± 0.38	0.21 ± 0.41	0.18 ± 0.39	0.13 ± 0.34	0.16 ± 0.37	0.741	0.005
HTN	0.24 ± 0.43 ^c^	0.35 ± 0.48 ^b^	0.53 ± 0.50 ^b^	0.61 ± 0.49 ^a^	0.63 ± 0.50 ^a^	0.001	0.069
HLD	0.31 ± 0.47	0.30 ± 0.46	0.33 ± 0.47	0.22 ± 0.42	0.32 ± 0.48	0.717	0.005
ARTH	0.32 ± 0.47	0.51 ± 0.50	0.54 ± 0.50	0.54 ± 0.50	0.58 ± 0.51	0.041	0.023
LBP	0.26 ± 0.44 ^d^	0.32 ± 0.47 ^bc^	0.54 ± 0.50 ^bc^	0.61 ± 0.49 ^b^	0.68 ± 0.48 ^a^	0.001	0.078

All data represent mean ± standard deviation. Symbols ^a, b, c,^ and ^d^ represent post hoc results from the Scheffe test. BMI, body mass index; OV, overweight; OB, obesity; DM, diabetes mellitus; HTN, hypertension; HLD, hyperlipidemia; ARTH, arthritis; LBP, low back pain.

**Table 2 medicina-59-01700-t002:** Age-dependent disparities in sarcopenic factors.

	Groups					*p*	η^2^
	Youngest-Old	Young-Old	Middle-Old	Old-Old	Oldest-Old		
	(*n* = 68)	(*n* = 168)	(*n* = 127)	(*n* = 46)	(*n* = 19)		
Calf circumference (cm)	29.44 ± 4.03	26.92 ± 3.18	24.66 ± 3.17	21.93 ± 3.46	19.90 ± 2.84	0.001	0.357
Grip strength (kg)	28.74 ± 5.97	22.89 ± 5.47	20.75 ± 5.13	16.39 ± 4.21	13.65 ± 2.05	0.001	0.348
5-time chair stand test (sec)	12.40 ± 1.63	16.43 ± 3.33	25.77 ± 6.04	30.65 ± 5.68	39.16 ± 4.30	0.001	0.723
ASM (kg/m^2^)	10.69 ± 2.05	7.84 ± 1.44	6.65 ± 1.16	6.67 ± 1.32	6.20 ± 0.51	0.001	0.488

All data represent mean ± standard deviation. ASM, appendicular skeletal muscle mass.

**Table 3 medicina-59-01700-t003:** Age-dependent disparities in lymphocyte subsets.

	Groups					*p*	η^2^
	Youngest-Old	Young-Old	Middle-Old	Old-Old	Oldest-Old		
	(*n* = 68)	(*n* = 168)	(*n* = 127)	(*n* = 46)	(*n* = 19)		
WBC (×10^3^ cells/μL)	5.99 ± 0.91	5.88 ± 0.71	5.71 ± 0.84	5.78 ± 0.82	5.80 ± 0.66	0.277	0.016
NK cells (%)	23.26 ± 4.67	17.92 ± 6.18	14.02 ± 5.12	10.33 ± 3.15	9.37 ± 3.02	0.001	0.371
CD3 T cells (%)	62.05 ± 8.76	59.52 ± 8.23	55.27 ± 7.16	53.74 ± 6.83	47.31 ± 7.46	0.001	0.170
CD4 T cells (%)	40.17 ± 7.60	38.94 ± 8.24	33.04 ± 6.56	27.93 ± 4.62	27.03 ± 3.65	0.001	0.274
CD8 T cells (%)	27.06 ± 6.41	22.07 ± 5.51	19.51 ± 4.46	15.68 ± 3.02	13.82 ± 1.28	0.001	0.328
CD19 B cells (%)	13.71 ± 2.83	13.18 ± 3.35	10.54 ± 2.78	9.56 ± 2.60	7.98 ± 2.07	0.001	0.253

All data represent mean ± standard deviation. WBC, white blood cells; NK, natural killer; CD, cluster differentiation.

## Data Availability

Not applicable.

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
