# Peer review of "Characteristics of Age Classification into Five-Year Intervals to Explain Sarcopenia and Immune Cells in Older Adults"

_medicina, 2023, doi:10.3390/medicina59101700_

Round 1

Reviewer 1 Report

Review

The paper entitled “Characteristics of Classification by Age for 5 Years to Explain Sarcopenia and Immune Cells in the Older Adults” deals with investigating the sarcopenic factors and immune cells in 433 older adults, divided into five age groups.

Results disclosed distinct alterations in both sarcopenia-related factors and immune cell functionalities among different age groups.

This manuscript makes a useful and valuable contribution to the knowledge and understanding of the subject matter.

Authors clearly showed that their research is replicable, and methodology was described in enough detail. The authors also followed the principles of good practice and ethical standards were maintained.

The obtained results are clearly explained, the conclusions follow a clear logic and are supported by adequate references. Considering that the population of 60+ year age group will reach 2.1 billion by the age 2050, it is clear that in order to have better health support for older general population, it is necessary to understand what are main problems that should be addressed.

I would have a few suggestions for some minor issues:

1.     Please, try to rephrase the title. This is not very understandable. As a suggestion, you may say “Characteristics of Age Classification by 5 Year intervals in Explaining Sarcopenia and Immune Cells in the Older Adults”.

2.     Line 10: Instead of “in living a same community” please use “in people living in the same community.”

3.     Line 11-13: There is no need to refer to age groups such as young-old, middle-old, etc. We are aware from the beginning that they are old people. Maybe it is enough to call the youngest/young/middle etc. However, I do not insist on this change.

4.     Line 37: I am not sure that I understand what you meant by the second part of this sentence: “Considering the diversity of health conditions and the increasing prevalence of chronic diseases, almost of research using 5-year age intervals could be suggested to be beneficial effects”. Please be more clear.

Author Response

Answers to reviewer’s comments 

Thank you for your kind advice and comments for publication in Medicina. We revised our manuscript as per your comments. We represented the specific modifications in response to the comments by blue-letters in my manuscript. We sincerely appreciate your comments because your comments make my manuscript better.

Comments and Suggestions for Authors

The paper entitled “Characteristics of Classification by Age for 5 Years to Explain Sarcopenia and Immune Cells in the Older Adults” deals with investigating the sarcopenic factors and immune cells in 433 older adults, divided into five age groups.

Results disclosed distinct alterations in both sarcopenia-related factors and immune cell functionalities among different age groups.

 This manuscript makes a useful and valuable contribution to the knowledge and understanding of the subject matter.

Authors clearly showed that their research is replicable, and methodology was described in enough detail. The authors also followed the principles of good practice and ethical standards were maintained.

The obtained results are clearly explained, the conclusions follow a clear logic and are supported by adequate references. Considering that the population of 60+ year age group will reach 2.1 billion by the age 2050, it is clear that in order to have better health support for older general population, it is necessary to understand what are main problems that should be addressed.

 I would have a few suggestions for some minor issues:

  1. Please, try to rephrase the title. This is not very understandable. As a suggestion, you may say “Characteristics of Age Classification by 5 Year intervals in Explaining Sarcopenia and Immune Cells in the Older Adults”.

#Response 1: Thank you for what the reviewer has pointed out the comment. I think it was awkward like your opinion, so I changed it. Ultimately, the revised title is as follows, and the text is also modified in the same way.

 “Characteristics of Age Classification by 5 Year intervals in Explaining Sarcopenia and Immune Cells in the Older Adults”

  1. Line 10: Instead of “in living a same community” please use “in people living in the same community.”

#Response 2: Thank you for what the reviewer has pointed out the comment. As per your comment, I change the phrase as below.

In lines 10 (New revised manuscript): “To achieve this, the study analyzed the variables related to sarcopenia and immune cells in people living in the same community.”

  1. Line 11-13: There is no need to refer to age groups such as young-old, middle-old, etc. We are aware from the beginning that they are old people. Maybe it is enough to call the youngest/young/middle etc. However, I do not insist on this change.

#Response 3: Thank you for what the reviewer has pointed out the comment. Certainly, I'm thankful for your understanding. The reason for dividing the elderly age group into 5-year intervals and naming them as such was to observe the changes in the human body due to aging in a more detailed manner. The detailed observation of aging can provide a more sensitive approach to factors that can affect health, such as immune cells, and it is believed that this could lead to the development of preventive measures for muscle loss syndrome.

  1. Line 37: I am not sure that I understand what you meant by the second part of this sentence: “Considering the diversity of health conditions and the increasing prevalence of chronic diseases, almost of research using 5-year age intervals could be suggested to be beneficial effects”. Please be more clear.

#Response 4: Thank you for what the reviewer has pointed out the comment. To help reviewers and readers understand, the above sentence has been modified as follows.

In lines 39 to 41 (New revised manuscript): “Given the wide range of health conditions and the rising prevalence of chronic diseases, it can be recommended that most research employing 5-year age intervals may yield beneficial outcomes.”

Thank you for your comments, I represented the modifications in response to your comments.

September 11, 2023

Reviewer 2 Report

I have reviewed the manuscript titled " Characteristics of Classification by Age for 5 Years to Explain Sarcopenia and Immune Cells in Older Adults." The manuscript presents a well-structured study investigating the relationship between age, sarcopenia-related factors, and immune cell functionality in older adults. The study contributes to our understanding of the age-related changes in muscle mass and immune function in older adults. The findings are consistent with previous research and highlight the importance of addressing sarcopenia and immune health in the aging population. However, there are some areas where the manuscript could be improved before publication.

It is worth noting that while the study confirms and aligns with previously established knowledge regarding age-related changes in muscle mass and immune function, it does not introduce entirely novel insights. Rather, it serves as a valuable confirmation of existing understanding in this field.

The study employs relatively straightforward statistical analyses to examine the relationship between age groups, muscle-related factors, and immune cell characteristics. While simplicity can be advantageous, it's important to acknowledge that more complex statistical methods or data modeling might provide a more nuanced understanding of the relationships under investigation.

It's crucial to acknowledge the limitation of a relatively small sample size in this study, which may impact the generalizability of the findings for reaching cut of values.

Suggestions for Improvement:

Abstract

The abstract briefly presents some key findings, such as the decline in calf circumference, grip strength, and natural killer cells with age, as well as variations in CD3 T cells, CD4 T cells, CD8 T cells, and CD19 B cells among different age groups. However, it lacks specific numerical data and statistical significance, making it difficult to assess the robustness of the results.

Introduction

In the introduction please clearly state what makes your study novel and different from existing research. What unique insights or contributions do you expect to make by categorizing participants into 5-year age groups? This will help set clear expectations for the reader to highlight the novelty.

Methods

Due to the description of the study design, a single-blind, randomized, prospective cohort study, Specify any randomization or blinding procedures in more detail if they are relevant to the study design.

Results

In Table 2, I did not understand what you mean by gender scores (1.35 ± 0.48 1.47 ± 0.50 1.51 ± 0.50 )

Consider providing a brief explanation of why certain clinical traits (e.g., hypertension, arthritis, low back pain) were selected for analysis, as this context can help readers understand the relevance of these factors to the study.

Please remove the limitation part from the conclusion to the end of the discussion. Ensure that the conclusion is concise and reinforces the main takeaways from the study.

Minor editing of English language required

Author Response

Answers to reviewer’s comments 

Thank you for your kind advice and comments for publication in Medicina. We revised our manuscript as per your comments. We represented the specific modifications in response to the comments by blue-letters in my manuscript. We sincerely appreciate your comments because your comments make my manuscript better.

Comments and Suggestions for Authors

I have reviewed the manuscript titled " Characteristics of Classification by Age for 5 Years to Explain Sarcopenia and Immune Cells in Older Adults." The manuscript presents a well-structured study investigating the relationship between age, sarcopenia-related factors, and immune cell functionality in older adults. The study contributes to our understanding of the age-related changes in muscle mass and immune function in older adults. The findings are consistent with previous research and highlight the importance of addressing sarcopenia and immune health in the aging population. However, there are some areas where the manuscript could be improved before publication.

It is worth noting that while the study confirms and aligns with previously established knowledge regarding age-related changes in muscle mass and immune function, it does not introduce entirely novel insights. Rather, it serves as a valuable confirmation of existing understanding in this field.

The study employs relatively straightforward statistical analyses to examine the relationship between age groups, muscle-related factors, and immune cell characteristics. While simplicity can be advantageous, it's important to acknowledge that more complex statistical methods or data modeling might provide a more nuanced understanding of the relationships under investigation.

It's crucial to acknowledge the limitation of a relatively small sample size in this study, which may impact the generalizability of the findings for reaching cut of values.

Suggestions for Improvement:

Abstract

The abstract briefly presents some key findings, such as the decline in calf circumference, grip strength, and natural killer cells with age, as well as variations in CD3 T cells, CD4 T cells, CD8 T cells, and CD19 B cells among different age groups. However, it lacks specific numerical data and statistical significance, making it difficult to assess the robustness of the results.

#Response 1: Thank you for what the reviewer has pointed out the comment. Based on your feedback, we calculated delta values between the five groups and provided statistical differences to increase the robustness of this study's results. The modifications are as follows, and the same modifications were made to the abstract and text.

New revised manuscript in lines 14 to 20: “With the progression of age, calf circumference (-8.4 to -11.05%; p = 0.001) and grip strength (-9.32 to -21.01%; p = 0.001) exhibited a noticeable reduction with each successive 5-year age bracket. Conversely, the capability to complete the 5-time chair stand demonstrated clear incline (32.49 to 56.81%; p = 0.001) starting from the middle-aged group. As for appendicular skeletal muscle mass, there was an evident tendency for it to decrease (-7.08 to -26.62%; p = 0.001) with increasing age. A gradual decline in natural killer cells became apparent within the old-old and oldest-old groups (-9.28 to -26.27%; p = 0.001).”

New revised manuscript in lines 191 to 195: “In addition, this study calculated delta values to assess differences between youngest-old and young-old groups, between young-old and middle-old groups, between middle-old and old-old groups, and between old-old and oldest-old groups, respectively. Furthermore, this study conducted an analysis using the Kruskal-Wallis test to examine the intergroup differences in these delta values.”

New revised manuscript in lines 253 to 257: “In details, the calf circumference showed differences between the youngest-old and young-old groups, with a decrease of -8.58%. Similarly, there was a -8.40% difference between the young-old and middle-old groups, an -11.05% difference between the middle-old and old-old groups, and a -9.27% difference between the old-old and oldest-old groups. These differences were statistically significant (Z = 140.957, p = 0.001, η² = 0.357).”

New revised manuscript in lines 258 to 262: “Grip strength exhibited variations across age groups: there was a -20.36% decline between the youngest-old and young-old groups, a -9.32% difference between the young-old and middle-old groups, a -21.01% difference between the middle-old and old-old groups, and a -16.72% difference between the old-old and oldest-old groups. These disparities were statistically significant (Z = 142.088, p = 0.001, η² = 0.348).”

New revised manuscript in lines 262 to 267: “The performance in the 5-time chair stand test displayed variations among age groups: there was a 32.49% increase between the youngest-old and young-old groups, a 56.81% difference between the young-old and middle-old groups, an 18.95% difference between the middle-old and old-old groups, and a 27.75% difference between the old-old and oldest-old groups. These differences were statistically significant (Z = 319.322, p = 0.001, η² = 0.723).”

New revised manuscript in lines 267 to 272: “The ASM exhibited disparities among age groups: there was a -26.62% decrease between the youngest-old and young-old groups, a -15.19% difference between the young-old and middle-old groups, a 0.33% difference between the middle-old and old-old groups, and a -7.08% difference between the old-old and oldest-old groups. These differences were statistically significant (Z = 176.718, p = 0.001, η² = 0.488).”

New revised manuscript in lines 294 to 298: “In details, the NK cells showed variations across age groups: there was a -22.95% decline between the youngest-old and young-old groups, a -21.79% difference between the young-old and middle-old groups, a -26.27% difference between the middle-old and old-old groups, and a -9.28% difference between the old-old and oldest-old groups. These differences were statistically significant (Z = 159.849, p = 0.001, η² = 0.371).”

New revised manuscript in lines 312 to 316: “In details, the CD3 T cells displayed differences across age groups: there was a -4.07% decrease between the youngest-old and young-old groups, a -7.13% difference between the young-old and middle-old groups, a -2.78% difference between the middle-old and old-old groups, and a -11.95% difference between the old-old and oldest-old groups. These disparities were statistically significant (Z = 68.006, p = 0.001, η² = 0.170).”

New revised manuscript in lines 316 to 321: “The CD4 T cells exhibited disparities among age groups: there was a -3.08% decrease between the youngest-old and young-old groups, a -15.15% difference between the young-old and middle-old groups, a -15.45% difference between the middle-old and old-old groups, and a -3.24% difference between the old-old and oldest-old groups. These differences were statistically significant (Z = 119.355, p = 0.001, η² = 0.274).”

New revised manuscript in lines 321 to 326: “The CD8 T cells displayed differences across age groups: there was an -18.43% decrease between the youngest-old and young-old groups, an -11.61% difference between the young-old and middle-old groups, a -19.64% difference between the middle-old and old-old groups, and an -11.83% difference between the old-old and oldest-old groups. These differences were statistically significant (Z = 135.165, p = 0.001, η² = 0.328).”

New revised manuscript in lines 326 to 331: “The CD19 B cells showed differences across age groups: there was a -3.89% decrease between the youngest-old and young-old groups, a -20.04% difference between the young-old and middle-old groups, a -9.26% difference between the middle-old and old-old groups, and a -16.55% difference between the old-old and oldest-old groups. These differences were statistically significant (Z = 112.429, p = 0.001, η² = 0.253).”

Introduction

In the introduction please clearly state what makes your study novel and different from existing research. What unique insights or contributions do you expect to make by categorizing participants into 5-year age groups? This will help set clear expectations for the reader to highlight the novelty.

#Response 2: Thank you for what the reviewer has pointed out the comment. By categorizing participants into 5-year age groups, this study's novelty lies in the early detection of muscle loss syndrome and immune cell function decline. This approach can pave the way for achieving a high quality of life and extending the lifespan. These concepts have been succinctly described in the introduction section as follows.

In lines 71 to 74 (New revised manuscript): “Such detailed age-specific categorization can make it easier to prevent muscle loss syndrome in the elderly and can be considered a proactive measure to prevent the decline in immune cell function that comes with aging. This can contribute to maintaining a high quality of life and extending lifespan.”

Methods

Due to the description of the study design, a single-blind, randomized, prospective cohort study, Specify any randomization or blinding procedures in more detail if they are relevant to the study design.

#Response 3: Thank you for what the reviewer has pointed out the comment. As per your comment, I inserted the sentences in the text in lines 81 to 86 (New revised manuscript).

“Through promotional activities at the Seoul Senior Tower, individuals who voluntarily enlisted in the study were expected to be between the ages of 60 to 100 and capable of independent living. Participants were unaware of the randomization results. The variables to be investigated in this study were determined at the time of analysis. The randomization list was generated by a computer using block randomization with five blocks.”

Results

In Table 2, I did not understand what you mean by gender scores (1.35 ± 0.48 1.47 ± 0.50 1.51 ± 0.50 )

#Response 4: Thank you for what the reviewer has pointed out. The "gender scores" presented in the results of this study are numerical values assigned during the analysis, where males were labeled as '1' and females as '2'. This was done to assess homogeneity between genders within each group. As per your comment, I inserted the sentences in the text in lines 205 to 208 (New revised manuscript).

“The "gender scores" presented in the results of this study are numerical values assigned during the analysis, where males were labeled as '1' and females as '2'. This was done to assess homogeneity between genders within each group.”

Consider providing a brief explanation of why certain clinical traits (e.g., hypertension, arthritis, low back pain) were selected for analysis, as this context can help readers understand the relevance of these factors to the study.

#Response 5: Thank you for what the reviewer has pointed out. The reason for investigating overweight, obesity, diabetes mellitus, hypertension, hyperlipidemia, arthritis, and low back pain was because these were the types of chronic conditions among the participants in this study. In other words, it was one of the foundational assessments aimed at excluding individuals with specific diseases in each group. While individuals in all five groups generally had chronic conditions, the goal was to differentiate whether there were any significant differences among them. As per your comment, I inserted the sentences in the text in lines 214 to 217 (New revised manuscript).

“The older adults who took part in this study commonly shared conditions such as overweight, obesity, diabetes mellitus, hypertension, hyperlipidemia, arthritis, and low back pain. When it comes to clinical traits, only hypertension, arthritis, and low back pain displayed noteworthy variations between the groups.”

Please remove the limitation part from the conclusion to the end of the discussion. Ensure that the conclusion is concise and reinforces the main takeaways from the study.

#Response 6: I appreciate the feedback provided by the reviewer. In response to your comments, I have made the necessary changes and included the sentences as follows in line 428 to 432 (New revised manuscript).

“In this study, when age was categorized in 5-year intervals, the researcher was able to observe a distinct pattern of muscle loss syndrome, and at the same time, the researcher confirmed a decline in the functions of immune cells. To prevent muscle loss syndrome in the elderly and observe declines in immune cell function, it is advisable to use 5-year age intervals, and through this approach, we can anticipate a higher quality of life.”

Thank you for your comments, I represented the modifications in response to your comments.

September 11, 2023

Reviewer 3 Report

Dear authors, I read your article with interest but I regret having to reject it for these reasons:

1) Overall, the information contained in the article is not innovative and indeed highlights concepts that are already well known in the literature.

2) The patient inclusion criteria are not clear and describing a geriatric patient as "an individual without particular physical health concerns" (line 84 in Materials and Methods) is in itself an error as we have the multidimensional geriatric assessment available which within this article it is never mentioned, both in terms of physical performance and cognitive performance.

3) The initial physical state of the patient and the amount of physical activity performed prior to the study is never mentioned. This parameter is fundamental because it defines the function of the subject and consequently it is not possible to classify subjects of the same age but with a different amount of physical activity carried out: a 60 year old marathon runner patient and an inactive 60 year old patient are two completely different subjects and not comparable.

4) The concept that the number of immune cells also overall defines the functionality of the immune system should be explored further.

Author Response

Answers to reviewer’s comments 

Thank you for your kind advice and comments for publication in Medicina. We revised our manuscript as per your comments. We represented the specific modifications in response to the comments by blue-letters in my manuscript. We sincerely appreciate your comments because your comments make my manuscript better.

Comments and Suggestions for Authors

I read your article with interest but I regret having to reject it for these reasons:

1) Overall, the information contained in the article is not innovative and indeed highlights concepts that are already well known in the literature.

#Response 1: Firstly, I would like to express my gratitude for your feedback. As you pointed out, there are indeed many papers similar to this manuscript. However, this study stands out as it specifically investigates elderly individuals living together in one place and sharing their daily lives, with a unique focus on dividing the age groups into 5-year intervals for a detailed examination of both muscle loss syndrome and immune cells in the elderly population. On the other hand, while creativity in the paper is important, in the context of research, "re" signifies "again," and "search" implies "seeking." Therefore, it is believed that true fruition is achieved through continuous "repeated seeking" or "repeated exploration." In response to the reviewer's suggestions, I have made extensive revisions from the introduction to the discussion to emphasize creativity as below, and I kindly request that you review it again.

In lines 39 to 41 (New revised manuscript): “Given the wide range of health conditions and the rising prevalence of chronic diseases, it can be recommended that most research employing 5-year age intervals may yield beneficial outcomes [5]. By using 5-year age intervals, researchers and healthcare practitioners can better account for the emergence of various chronic conditions and the potential differences in health status within specific age ranges.”

In lines 71 to 74 (New revised manuscript): “Such detailed age-specific categorization can make it easier to prevent muscle loss syndrome in the elderly and can be considered a proactive measure to prevent the decline in immune cell function that comes with aging. This can contribute to maintaining a high quality of life and extending lifespan.”

2) The patient inclusion criteria are not clear and describing a geriatric patient as "an individual without particular physical health concerns" (line 84 in Materials and Methods) is in itself an error as we have the multidimensional geriatric assessment available which within this article it is never mentioned, both in terms of physical performance and cognitive performance.

#Response 2: I apologize for the confusion in the manuscript content. I have made the following revisions as per your suggestions, and the text has been updated accordingly.

In lines 94 to 96 (New revised manuscript): The criteria for inclusion encompassed individuals without particular physical health concerns, a certain level of physical activity, or an absence of physical activity-related problems, and a lack of recent surgeries within the last three months. → “The inclusion criteria comprised individuals without cancer or severe illnesses, those capable of performing a certain level of physical activity, and individuals who had not undergone surgery within the past 3 months.”

3) The initial physical state of the patient and the amount of physical activity performed prior to the study is never mentioned. This parameter is fundamental because it defines the function of the subject and consequently it is not possible to classify subjects of the same age but with a different amount of physical activity carried out: a 60 year old marathon runner patient and an inactive 60 year old patient are two completely different subjects and not comparable.

#Response 3: Thank you for what the reviewer has pointed out the comment. The daily physical activity levels of the pre-screened participants were initially assessed but were considered unnecessary and were not included in the manuscript. However, following the reviewer's advice, the following information has been included in the text.

In lines 112 to 116 (New revised manuscript): “The participants' daily physical activity levels were assessed and quantified using the International Physical Activity Questionnaire (IPAQ) - shortened form version (Cheng, 2016). Participants filled out questionnaires based on their weekly physical activity records, and daily calorie expenditure was determined by calculating metabolic equivalent-minutes.”

In lines 224 to 230 (New revised manuscript): “In terms of daily calorie expenditure from physical activities, the average calories burned through these activities were as follows: 1740.35 ± 289.28 MET⋅min/week for the youngest-old group, 1627.90 ± 233.20 MET⋅min/week for the young-old group, 1343.36 ± 187.45 MET⋅min/week for the middle-old group, 1211.28 ± 289.28 MET⋅min/week for the old-old group, and 1132.58 ± 208.81 MET⋅min/week for the oldest-old group. Significant differences were observed among these groups (X² = 181.708; p = 0.001; η² = 0.422).”

In lines 341 to 342 (New revised manuscript): “Furthermore, it was verified that the daily physical activity levels of the elderly progressively decline with advancing age.”

In lines 420 to 423 (New revised manuscript): “Eventually, this study discovered that with advancing age, muscle mass decreases, evaluation parameters related to muscle loss factors worsen, and consequently, immune cell function deteriorates. Furthermore, it was determined that these findings were more pronounced when age was categorized into 5-year intervals.”

Reference: [14] Cheng, H.L. A simple, easy-to-use spreadsheet for automatic scoring of the International Physical Activity Questionnaire (IPAQ) Short Form (updated November 2016). ResearchGate, 2016.

4) The concept that the number of immune cells also overall defines the functionality of the immune system should be explored further.

#Response 4: Thank you for what the reviewer has pointed out the comment. As per the reviewer's feedback, the explanation regarding the function of immune cells has been elaborated upon in lines 160.

In lines 165 to 168 (New revised manuscript): “Natural killer cells have the ability to identify infected or cancerous cells, binding to them, and then releasing enzymes and other substances that effectively break down the outer membrane of these cells (Park et al., 2023). These cells play a significant role in the early defense against viral infections.”

In lines 168 to 174 (New revised manuscript): “At birth, acquired immunity is not present; instead, it develops through a learning process. This learning process initiates when an individual's immune system encounters a foreign intruder and identifies its antigen. Subsequently, the adaptive immune components learn the most effective way to combat this antigen and commence building a memory for it (Jee, 2020). Acquired immunity is also known as specific immunity because it orchestrates a tailored response against specific antigens encountered in the past.”

In lines 174 to 178 (New revised manuscript): “As we grow older, our immune system tends to lose some of its ability to differentiate between self and non-self. Consequently, autoimmune diseases may manifest more frequently. For instance, the quantity of white blood cells and their subsets capable of responding to new antigens diminishes, which in turn diminishes the body's capacity to retain memory and mount a defense when encountering new antigens (Franceschi et al., 2018).”

References:

[18] Park, S.; Park, S.K.; Jee, Y.S. Moderate- to fast-walking improves immunocytes through a positive change of muscle contractility in old women: a pilot study. J Exerc Rehabil. 2023, 19(1), 45-56. doi:10.12965/jer.2244512.256

[19] Jee, Y.S. Influences of acute and/or chronic exercise on human immunity: third series of scientific evidence. J Exerc Rehabil. 2020, 16(3), 205-206. doi:10.12965/jer.2040414.207

[20] Franceschi, C.; Garagnani, P.; Parini, P.; Giuliani, C.; Santoro, A. Inflammaging: a new immune-metabolic viewpoint for age-related diseases. Nat Rev Endocrinol. 2018, 14(10), 576-590. doi:10.1038/s41574-018-0059-4

Thank you for your comments, I represented the modifications in response to your comments.

September 11, 2023

Round 2

Reviewer 2 Report

“The "gender scores" presented in the results of this study should be removed.  the information in the article is not particularly original and really emphasizes ideas that are already well-known in the literature. However, they have addressed my other concerns

minor language editing is necessary

Author Response

Dear Reviewer

 “The "gender scores" presented in the results of this study should be removed.  the information in the article is not particularly original and really emphasizes ideas that are already well-known in the literature. However, they have addressed my other concerns.

Response 1: Thanks for your suggestion. According to your point, we removed "gender scores" in the results. Thank you so much again.

minor language editing is necessary

Response 2: Thanks for your suggestion. Based on your opinion, we requested English correction.

Reviewer 3 Report

Overall there is a good improvement  of the manuscript 

Author Response

Dear Reviewer

Overall there is a good improvement of the manuscript 

Response 1: Thank you for all your comments and teachings.